# The Impact of RSV Hospitalization on Children’s Quality of Life

**DOI:** 10.3390/diseases11030111

**Published:** 2023-08-30

**Authors:** August Wrotek, Oliwia Wrotek, Teresa Jackowska

**Affiliations:** 1Department of Pediatrics, Centre of Postgraduate Medical Education, 99/103 Marymoncka Str., 01-813 Warsaw, Poland; tjackowska@cmkp.edu.pl; 2Department of Pediatrics, Bielanski Hospital, 80 Cegłowska Str., 01-809 Warsaw, Poland; 3Student Research Group, Bielanski Hospital, 80 Cegłowska Str., 01-809 Warsaw, Poland

**Keywords:** respiratory syncytial virus, bronchiolitis, pneumonia, burden, caregiver, infant, quality-adjusted life years (QALYs), health-related quality of life, cost-effectiveness

## Abstract

Background: Respiratory syncytial virus (RSV) is one of the most frequent etiological factors of lower respiratory tract infections in children, potentially affecting patients’ quality of life (QoL). We aimed to asses QoL in children under 2 years of age hospitalized due to laboratory-confirmed RSV infection. Methods: A QoL was assessed by parents/tutors with the use of the 100-point visual analog scale and compared against a disease-free period. We evaluated the median utility, QoL loss (reported in days), and quality-adjusted life years (QALY) loss in relation to RSV hospitalization. Results: We included 132 patients aged from 17 days to 24 months (median 3.8 months). The mean utility during the hospitalization varied between 0.418 and 0.952, with a median of 0.679 (95%CI: 0.6–0.757) and median loss of 0.321 [0.243–0.4], which further translated into a loss of 2.2 days (95%CI: 1.6–3.1). The QALY loss varied between 0.526 × 10^−3^ and 24.658 × 10^−3^, with a median of 6.03 × 10^−3^ (95%CI: 4.38–8.48 × 10^−3^). Based upon the final diagnoses, the highest QALY loss was 6.99 × 10^−3^ (95%CI: 5.29–13.7 × 10^−3^) for pneumonia, followed by bronchiolitis—5.96 × 10^−3^ (4.25–8.41 × 10^−3^) and bronchitis—4.92 × 10^−3^ (2.93–6.03 × 10^−3^); significant differences were observed only between bronchitis and pneumonia (*p* = 0.0171); the QALY loss was not age-dependent. Although an increasing tendency in the utility score was observed, a strong cumulative effect related to the length of stay was noted until day 13. Conclusions: RSV contributes significantly to the utility deterioration and QALY loss in the case of RSV hospitalization, and the patient-reported data should be used in pharmacoeconomic assessments of the impact of RSV.

## 1. Introduction

Respiratory syncytial virus (RSV) is one of the most significant infectious agents, re-sponsible for around 33 million episodes of lower respiratory infection (LRTI) per year and approximately 3.6 million hospitalizations in children under 5 years of age [1]. The global estimates associate RSV with approximately 100,000 pediatric deaths in this age group, including over 26,000 in-hospital deaths [1]. The most severe disease course is observed mainly in the youngest group of patients, with 6.6 million LRTI episodes and 1.4 million hospitalizations in infants aged 0–6 months [1]. The total impact of RSV in children is of extreme importance; it is estimated that 2% of all deaths in children aged 0–6 months and 3.6% in those aged from 28 days to 6 months are related to RSV; the vast majority of fatal cases occur in low- and middle-income countries [1]. Although hospitalization rates decrease with age, it is estimated that even 687 hospitalizations per 100,000 children under 2 years old occur each year in high-income countries [2]. Moreover, RSV infections are not only limited to the youngest patients or pediatric population in general, but also present a significant problem in adult patients, with the elderly as one of the risk groups of a severe disease course [3,4,5].

Cost-effectiveness analyses are crucial for regional and global healthcare policy-makers, as potential decisions regarding the implementation of any novel diagnostic, therapeutic, or preventive measure are based upon a thorough cost–utility assessment. Currently, a number of studies are being conducted in order to improve anti-RSV prophylaxis methods (focusing on both monoclonal antibodies and vaccines) as well as antiviral treatment [6,7,8,9,10,11]. The cost-effectiveness analyses initially focused on the costs from a payer’s perspective; however, except for the clinical outcome measures, which include mainly mortality and the frequency of complications, which are major cost-drivers, the awareness of the significance of patient-reported outcome measures (PROMs) and patient-reported experience measures (PREMs) is growing [12]. A societal perspective includes the above-mentioned points of view, as they also influence patients’ utility and productivity. There are two groups of health-related quality of life (HRQoL) assessment methods, direct and indirect, and the most commonly used systems include the visual analogue scale (VAS), time trade-off scale, and standard gamble among the direct methods, and Eu-roQol-5-Dimension (EQ-5D), health utilities index, and short form-6 dimension (SF-6D) among the indirect methods [13]. The importance of PROMs is being stressed and its wide range of applications has been suggested for various purposes, from the end-point in clinical trials to the assessment of the healthcare system [12,14,15]; nevertheless, there exist few reported methods of PROM assessment in children, especially in the youngest group of patients [14,16,17].

It needs to be emphasized that, although the duration of the RSV disease is relatively short, the impact of RSV on the patient’s quality of life is significant, as the disease provokes disturbing signs/symptoms (cough, fever, and dyspnea, for example); affects feeding; and causes listlessness, malaise, and/or general discomfort [17,18]. An in-depth knowledge of the effects of RSV on both clinical and HRQoL outcomes is needed. Scarce da-ta have been published with regard to HRQoL in the course of an RSV infection, yet the studies showed a severe impairment in quality of life (reaching 40% during the first week of the disease) [19], with a more severe impact on children younger than 5 years old (compared with older patients and adults) and for those who seek (versus those who do not seek) health care [20]. In addition, emphasis has been placed on the influence of the RSV-caused illness on the HRQoL of parents/caregivers and families [17,19,21]. The estimates of the RSV-related loss of QALY (quality-adjusted life years) report values between 3.82 × 10^−3^ and 16.9 × 10^−3^ in children under 5 years of age [20,22].

Data on HRQoL during an RSV disease are highly needed; therefore, we performed this prospective study on the quality of life of children under 2 years old who were hospitalized with an RSV infection.

## 2. Materials and Methods

This prospective study included patients under 2 years of age hospitalized due to a laboratory-confirmed RSV infection in the period from 28 December 2018 to 31 March 2020 at the Bielanski Hospital, Warsaw.

The inclusion criteria consisted of respiratory tract infection signs/symptoms (runny nose, cough, dyspnea, increased body temperature, abnormalities in the physical examination according to the definitions presented below); asymptomatic carriers were not included, as they might be expected to be less disturbed by RSV and to not reflect the impact of the disease (Table 1). Only patients with community-acquired infections were eligible, because a nosocomial origin of the infection might aggravate the parental perception of the disease severity and lead to a lower HRQoL evaluation. The exclusion criteria consisted of a nosocomial origin of the infection; the lack of full knowledge on the clinical course; or lost to follow-up, i.e., the lack of a baseline quality of life (QoL) assessment in a disease-free period. The definitions were based upon the Polish guidelines on the management and treatment of respiratory tract infections [23].

### 2.1. Laboratory Procedures and Clinical Definitions

The etiology of the infection was confirmed with a positive result of either the rapid antigen test or the reverse transcription polymerase chain reaction (RT-PCR). While Alere Binax NOW (Alere Scarborough Inc.; Scarborough, ME, USA) was the antigen test of choice, the choice of RT-PCR was RSV Xpert Xpress Flu/RSV GeneXpert (manufactured by Cepheid, Sunnyvale, CA, USA); in the case of any discrepancies, RT-PCR was considered conclusive. The samples were taken from nasopharyngeal swabs, and the analyses were performed according to the manufacturers’ instructions.

The final discharge diagnosis coded according to the 10th revision of International Classification of Diseases (ICD-10) was used to assign patients to subgroups of pneumonia, bronchitis or bronchiolitis with codes J12.1, J20.5, and J21.0, respectively (Table 1) [23].

Another subgroup analysis was based upon the age on admission, including four major age groups: <3 months old, 3–5 months old, 6–11 months old, and 12–23 months old. The choice of the age subgroups was motivated by the known age-related risk groups used in Poland for the assessment of the need for hospital treatment (i.e., <3 months old for bronchiolitis and <6 months old for pneumonia) [23]. Additionally, an analysis in a month-by-month age division was also performed.

The primary end-points were as follows: the loss of utility during an RSV-caused hospitalization, a HRQoL loss (expressed in days) during the whole hospital treatment period, and a total QALY loss during hospitalization. The secondary end-points were an analysis on the utility changes on a day-by-day basis and a comparison of the HRQoL loss with regard to the different sites of infection, i.e., bronchiolitis, bronchitis, and pneumonia.

### 2.2. HRQoL Assessment and Calculations

A proxy perspective is expected to be the most reliable for the purposes of HRQoL evaluation; therefore, every parent/legal tutor who was taking care of a patient during the hospital stay was asked to indicate changes in the HRQoL. As there is no validated scale or scoring system to assess the HRQoL in children under 5 years of age, a visual analogue scale (VAS) was chosen. The influence of the disease on the HRQoL was assessed during the hospitalization period only, thus decreasing the effect of potentially confounding factors, such as socioeconomic variation, family support, and so on. The parents/tutors were asked to indicate, on a 100-point VAS, the quality of life on the consecutive days of hospitalization, with 100 points meaning “the best imaginable quality of life” and 0 referring to death. Furthermore, parents were contacted at least 1 week after hospital discharge and asked about the quality of life at a disease-free time point.

The utility on the consecutive days was obtained from the parents/tutors, and the total utility during the hospitalization was calculated as a mean (i.e., the sum of particular utilities divided by the number of in-hospital days) and presented as a fraction (between 0 and 1). The utility loss during the whole hospitalization was calculated by multiplying the mean by the median number of days (length of stay) and by the kappa coefficient; the kappa coefficient (between 0 and 1) corresponded to the baseline QoL assessed by the parent/tutor during a period free from diseases.

QALY calculations:

To calculate the QALY, the quality of life during the hospitalization was adjusted for the baseline utility in the disease-free period in the following manner:A = QoL outside the hospitalization = (365 − *l*) × kappa
B = QoL during the hospitalization = *l* × *util*
C = aggregated QoL in the year when the RSV episode took place= [A] + [B]= (365 − *l*) × kappa + *l* × *util*
D = expected QALY = 365 × kappa
E = QoL loss [days] = [D] − [C] = 365 × kappa − [(365 − *l*) × kappa + *l* × *util*]
F = QALY loss = [E]/[D] = [365 × kappa − [(365 − *l*) × kappa + *l* × *util*]]/365 × kappa
where *l*—length of stay (days) and *util*—mean utility during the hospitalization (calculated by dividing the sum of utilities on the hospitalization days by the number of hospitalization days), while kappa corresponds to the baseline QoL assessed by a parent/tutor in a period free from disease.

First, the QALY loss on specific days was calculated for each patient, and then the QALY loss on consecutive lag periods was evaluated by adding the QALY loss on separate days (i.e., QALY loss for days 1 + 2, 1 + 2 + 3, 1 + 2 + 3 + 4, and so on, up to days 1 + … + 16). Each patient’s data were analyzed separately, and both the mean or median (depending on the data distribution) QALY loss on consecutive days for the whole group and the mean or median cumulative QALY loss over the above time periods were presented.

### 2.3. Statistical Analysis

The Shapiro–Wilk test was performed for the assessment of data distribution, and normally distributed data were further presented as the mean and standard deviation (SD), while the median value and the interquartile range (IQR) were used for skewed data. A corresponding parametric or nonparametric test (Student’s *t*-test or Mann–Whitney U test, respectively) was used for comparisons of the two groups, while for the comparisons of multiple groups, one-way ANOVA (with Bonferroni correction) or the non-parametric Kruskal–Wallis test (with a multiple rank comparison) was performed. The contingency table and two-way Chi square tests were employed to compare the categorical variables. Spearman’s rank correlation test was used to verify the relationship between the utilities and consecutive days of hospital stay and a correlation coefficient (*rho*) was calculated. Statistical significance was defined as a *p*-value of less than 0.05. The statistical analysis was carried out using the Statistica 13.1 software package (Statsoft, Tulsa, OK, USA).

The study received appropriate ethical approval from the local ethics committee at the Centre for Postgraduate Medical Education, Warsaw (approval number 115/PB/2018, issued on 7 November 2018). The study was conducted in accordance with the Declaration of Helsinki with its subsequent amendments. An informed consent was received from parents/tutors prior to the study enrollment.

## 3. Results

In total, during the analyzed period, there were 250 laboratory-confirmed RSV hospitalizations. Parents/tutors of 183 children were asked to take part in the study, and an informed consent was obtained from 147 patients, while fully completed questionnaires were obtained from 142 patients; in 10 cases, the attempts to follow-up (in order to establish the baseline QoL) failed. Finally, 132 children formed the study group (Figure 1). Patients were aged 10 days to 720 days, with a median age of 3.8 months (IQR: 2.05–6.88 months). The length of stay varied between 3 and 16 days (median 7 days). The median baseline utility in the disease-free period (the kappa coefficient) reached 1, varying between 0.9 and 1. The majority of the patients were diagnosed with bronchiolitis (n = 100), followed by pneumonia (n = 22) and bronchitis (n = 10), and significant differences were observed with regard to the age (median 3.13, 7.03, and 15.4 months, respectively, *p* < 0.01), but not with regard to the length of stay (*p* = 0.3382). The baseline characteristics of the study group are shown in Table 2.

The utility assessed by the parents/tutors varied between 0.1 and 1 and the mean utility during the hospitalization varied between 0.418 and 0.952, with a median of 0.679 (95%CI: 0.6–0.757).

We observed an increasing tendency in the utility score in the whole group, as well as in the subgroup division based on the length of stay, although a small decrease was observed in the whole study group between days 9 and 10, as well as days 12 and 13 (Figure 2). Spearman’s rank correlation test revealed a high correlation between the median utility and consecutive days, with a correlation coefficient of 0.935 (Figure 3). The utility loss reached a median value of 0.321 (95%CI: 0.243–0.4), which further translated into a loss of 2.2 days (95%CI: 1.6–3.1).

The QALY loss varied between 0.526 × 10^−3^ and 24.658 × 10^−3^, with a median RSV-attributable QALY loss of 6.03 × 10^−3^ (95%CI: 4.38–8.48 × 10^−3^). With regard to the diagnoses, the highest QALY loss was observed in the case of pneumonia (6.99 × 10^−3^, 95%CI: 5.29–13.7 × 10^−3^), followed by bronchiolitis (5.96 × 10^−3^, 95%CI: 4.25–8.41 × 10^−3^) and bronchitis (4.92 × 10^−3^, 95%CI: 2.93–6.03 × 10^−3^); however, statistically significant differences were observed only between bronchitis and pneumonia (*p* = 0.0171), and they were not related to the differences in the length of stay, which remained insignificant (Figure 4). 

Whereas the differences in QALY loss were related to the final diagnosis, we observed no differences regarding the patients’ age; the Kruskal–Wallis test showed no statistically significant differences between the major age groups (i.e., under 3 months old, 3–5 months old, 6–11 months old, and 12–23 months old), nor between the age groups analyzed month by month (Figure 5).

The increasing length of stay resulted in a higher QALY loss (Figure 6), and a cumulative effect on QALY loss was observed until day 13 (Figure 7).

## 4. Discussion

The question of RSV-related quality of life is fundamental to the pharmacoeconomic assessment of the prevention strategies or treatment technologies in RSV infections. Despite its importance, there remains a huge paucity of evidence on RSV-related QoL in children, and the issue seems to be underestimated. For the moment, to the best of our knowledge, this study is one of the very few research attempts performed directly regarding RSV-infected patients.

Our investigation shows that RSV significantly contributes to the quality of life loss in the case of hospitalization.

Although there is a limited number of previously published studies, our results expand the knowledge and further confirm the observed association between RSV hospitalization and a significant QoL loss. A recent study by Diez-Gandia prospectively enrolled 86 children under the age of 2 with a laboratory-confirmed RSV infection; the patients were recruited mostly from primary care (eight sites), with from one hospital [19]. A newly ad hoc developed 38-item questionnaire was used at four timepoints (days 0, 7, 14, and 30) to evaluate the HRQoL. The study revealed a median loss in QoL of 37.5% and 31.5% on day 0 and 7, respectively. Day 0 and 7 refer to days 1 and 8 in our study, and a direct comparison in terms of the utility shows a more severe utility loss on the first day of assessment in our study and less severe loss after a week (50% vs. 37.5% and 10% vs. 31.5%, respectively). A plausible explanation for those discrepancies includes the differences between the groups—children who are hospitalized are affected to a higher degree than those in primary care, and it is reflected in more severe HRQoL loss; secondly, because the symptoms of RSV infection (such as in the case of bronchiolitis, which is a typical model of RSV infection) usually peak between days 3 and 5 [24], the observed relatively slow pace of improvement in the study by Diez-Gandia might be explained by the fact that the disease severity and the patients’ HRQoL might have initially dropped (at day 3–5) to start restoring on the following days, while the hospitalized patients were admitted around their peak utility loss.

A systematic review by Glaser presented studies from the USA that assessed the utilities in children under 60 months of age (together with the utilities in caregivers). The review of two of these studies included original cohort studies that calculated the HRQoL loss in premature infants, and the net QALY loss calculated by the reviewers revealed the value of 16.9 × 10^−3^ QALY per nonfatal RSV episode from the onset to 60 days [17]. It needs to be remembered that patients under 5 years of age were included in the review, while we only enrolled patients under 2 years old. A study by Hodgson showed that the quality of life is affected more seriously in younger children and, although the study compared children over and under 5 years of age, a similar association might be expected in younger age groups, because the vast majority of hospitalizations and a severe disease course is observed in the youngest children [20]. However, although the assumption of the review was to include patients under 5 years of age, in fact, the above results are based on two original studies by Pokrzywinski and Leidy, who enrolled patients under the age of 12 months and 30 months (212 and 46 children), respectively, thus age-related differences might be expected [25,26]. While a more significant QoL loss might have been anticipated in the youngest group of patients, we observed no differences in terms of QoL in the age subgroups in our study; this fact emphasizes the magnitude of the impact of RSV hospitalization on the patients’ quality of life.

Our research focused on the whole pediatric population, whereas children in the above studies had a history of prematurity; nonetheless, a review by Glaser presented results already adjusted for prematurity [17]. In our group of patients, we did not correct the results for gestational age, as the study protocol assumed the assessment in a regular cohort of patients who are referred to the hospital; additionally, our follow-up evaluation of the baseline QoL in the disease-free period diminishes the influence of any chronic health condition.

Noticeably, we observed a lower QALY loss compared with the review by Glaser [17], but the period evaluated in that review was much longer than in our study, which might offer a credible explanation. On the one hand, the duration of the assessment should not be overestimated, as signs/symptoms of the RSV infection and the related QoL loss are expected to be observed mainly in the acute period; a study by Diez-Gandia assumed day 30 to be the baseline day and compared all the results against it, showing that, at day 14 of the disease, the utility loss drops to 8.9% [19]. On the other hand, the duration of the RSV-related utility decrease should be remembered, while our study offers the results from the hospitalization period appraisal only; thus, it should be treated as a minimal reflection of the RSV-related QoL disturbances. The above studies showed that an RSV hospitalization affects children’s QoL for up to 30 [25] or even 60 days post-discharge [26]; in our group of patients, the median baseline kappa coefficient was 1 and all of the reported baseline utilities were high, but we did not evaluate the length of the period for which the RSV hospitalization affected the HRQoL, focusing on the utilities reported during the hospital stay. As long as the magnitude of the medium-term influence of an RSV-hospitalization seems to be rather low, it needs to be emphasized that the repercussions of the RSV hospitalization might reach far beyond the acute phase, and especially beyond the hospitalization period.

Another interesting phenomenon that needs to be discussed is a general increase in the HRQoL (with short episodes of a plateau) until day 9, with a subsequent decrease in the HRQoL observed on day 10 (a similar decrease was observed between days 12 and 13); such decreases were observed in subgroup analyses as well. Nonetheless, those short decreases are probably due to individual variabilities and, especially, general graphs are prone to the bias, as they include patients at different stages of the disease, including those who worsened during the hospital stay. In general, a strong increasing tendency in HRQoL was observed during the hospitalization period.

We also observed a strong cumulative effect, in line with the previous studies, including the meta-analysis by Glaser, who observed a steep rise in the first part of the episode and a more moderate rise in the second part; because the data were derived from preterm infant studies, the authors postulated the need for the assessment in term infants [17]. In our study, we observed a continuous rise until day 13, with a following slight decrease in the median QALY loss; however, methodological differences have to be remembered; while our data were gathered only during the hospitalization and the number of patients on the consecutive days was decreasing, the meta-analysis shows a day-by-day assessment for a predefined period of time and is based on comparable numbers of patients on each day. Thus, individual characteristics of the patients may have had an impact on the less-expressed cumulative effect, as those hospitalized for a longer period of time may not have exhibited such a high total QALY loss; however, the graph shows that, in the case of our patients, the QALY loss lasted at least until day 15.

Another estimation of HRQoL for those under the age of 5 was conducted by Hodgson and colleagues, who used a model for the estimation of the peak HRQoL loss, based on the results received from older children; the calculated QALY loss varied between 3.024 and 3.823 × 10^−3^ in those who did not seek and those who sought health care, respectively [20]. The quality of life was disturbed to a lesser extent than in our group of patients, yet, as stated before, it was based upon the results from ambulatory patients, and did not derive from a direct assessment of the patient’s HRQoL, but from a regression model. To evaluate the influence of the RSV on HRQoL, we chose a real-time direct assessment by proxy with the use of a VAS. The studies using an approach based upon a numerical rating from the caregivers are preferred and, for instance, obtained the highest-quality assessment score in the systematic review by Glaser [17]. Nevertheless, some attempts have been made to create other tools that would more specifically reflect RSV infections, such as the questionnaire developed by Diez-Gandia [19]. Another proposed option is the use of Pediatric Quality of Life Inventory (PedsQL)™ 4.0 generic core scales for those aged 2–4 years and PedsQL™ 4.0 infant scales, with a separate version for 1–12-month-olds and 13–24-month-olds; the protocols have been positively verified in a group of RSV-infected children, although some difficulties with the HRQoL scoring in the case of young children were observed [27].

The burden of RSV is enormous; a study by Villamil et al. intended to estimate the direct effect of RSV disease from a national perspective in Colombia—the authors computed a total loss of 260,873 years of life in Colombian children under 2 years of age in 2019, driven by RSV bronchiolitis only [18]. The authors concluded that a mean DALY (i.e., disability-adjusted life years) rate reached 20 DALYs/1000 person-year (95%CI: 16–27) [18]. It has to be remembered, however, that the study used only the most important health outcomes, which included the presence or absence of complications (such as hypoxemia or pneumonia), including severe complications (ICU admission and sepsis) as well as long-term complications (recurrent wheezing) [18]. While the latter might raise some polemics, it is being widely investigated and we are gaining more insights into the pathophysiological mechanisms involved and the underlying circumstances that increase the risk of post-viral wheezing; moreover, different risk assessment results might derive from syndromes overlapping between various phenotypical bronchiolitis subgroups [28,29,30]. While the medical aspect of RSV infections is widely known, we intended to highlight how RSV influences the patients’ quality of life, as well as a vast spectrum of effects, including long-term complications, but the caregivers’ stress or loss of work productivity also needs to be understood, as it all translates into high health care resource utilization [31].

There exist certain strengths and limitations to this study. Firstly, any generalizations need to be made with precaution, as this is a single-center study, and the evaluation was performed by parents/tutors; thus, important differences regarding the disease perception and expectations of the health status or the patient’s quality of life might derive from particular population-related variations. This problem has been widely described, and different weights can be locally assigned to various utility statuses, mostly in the case of the studies based upon QoL scales/scores. Nevertheless, the disease itself is a global concern and has a repeatable clinical course; therefore, the repercussions on HRQoL can be expected to be similar. The use of the visual analogue scale—an easily comprehensible, universally used HRQoL assessment tool—diminishes the influence of linguistic skills, educational level, or cognitive abilities. Although the use of any particular tool may distort the results, it needs to be expressed clearly that a direct HRQoL assessment is impossible at this age, and no widely used or verified HRQoL scales exist. Certainly, several factors could have biased the proxy assessment of HRQoL, such as personal situation, level of family support, number of people per room, behavior of hospital roommates, previous experiences with medical care, especially hospitalization experiences, or level of anxiety. This study is unable to encompass the entire problem of the RSV-related QoL loss because of the relatively small sample size and the period of evaluation that lasted the hospitalization time only; on the other hand, a day-by-day assessment is preferred and mirrors the true effects on the HRQoL more precisely than post-factum or en bloc assessment. Although the duration of the post-hospitalization QoL loss might raise some controversies, it should not be forgotten or neglected, and future studies assessing QoL loss in the post-hospital period are highly needed as well. Notwithstanding these limitations, the data on this topic are scarce and our work offers valuable insights into the RSV-related quality of life loss.

The practical implications of this study go beyond simple calculations of utility or QALY loss; firstly, it must be recognized that hospitalization for RSV has a significant impact on quality of life; secondly, the most important question from the patient’s perspective is whether there are ways to improve quality of life during hospitalization. There are a number of interventions that could be considered, ranging from psychological support (both from family/relatives and hospital staff, preferably with the help of a team of psychologists) to minor effects of the medications used during hospitalization. The question of support leads to another issue—support for caregivers, who are also exposed to severe stress and suffer from a reduction in their quality of life, which shows the importance of social conditions in pediatric wards. We argue that the patient’s perspective is as important as the payer’s perspective, and that pharmacoeconomic evaluation of health technologies should include both. The impact of RSV extends beyond hospital medicine, and it is essential to assess RSV-related loss of quality of life in the ambulatory setting in order to recognize the true burden of the disease.

## 5. Conclusions

In conclusion, our study clearly shows a significant contribution of RSV to the deterioration in utility and QALY loss in the case of hospitalization; we stipulate that patient-reported data (or, as in this case, reported by the patient’s caregiver) should be used in pharmacoeconomic assessments of the impact of RSV.

## Figures and Tables

**Figure 1 diseases-11-00111-f001:**
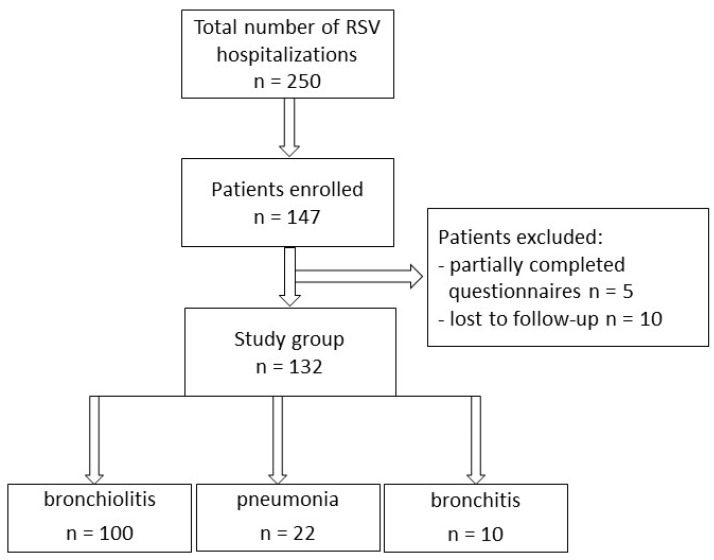
Flowchart of the patients in the study.

**Figure 2 diseases-11-00111-f002:**
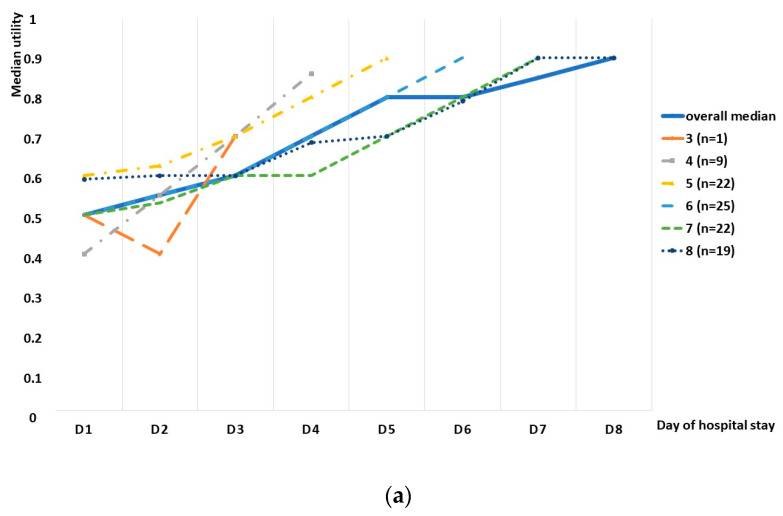
Tendencies in utilities regarding the length of stay. The graphs show median utilities with regard to the length of hospital stay: (**a**) in children hospitalized for the period of 3–8 days and (**b**) in those hospitalized for 9–16 days. A solid blue line indicates the median utilities on the consecutive days, calculated for the whole study group, while other colors refer to particular length of stay groups and show utilities on the particular days (D1, D2, …, D16 correspond to day 1, day 2, …, day 16).

**Figure 3 diseases-11-00111-f003:**
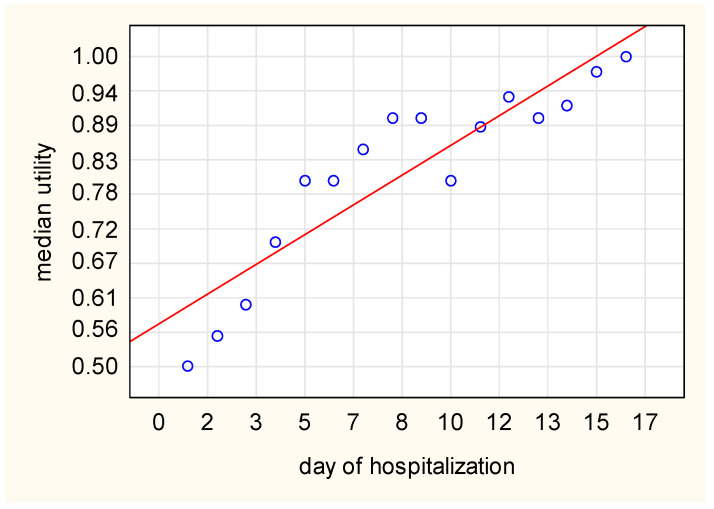
Correlation between the median utility and consecutive days of hospitalization (based on Spearman’s rank correlation test, *rho* = 0.935).

**Figure 4 diseases-11-00111-f004:**
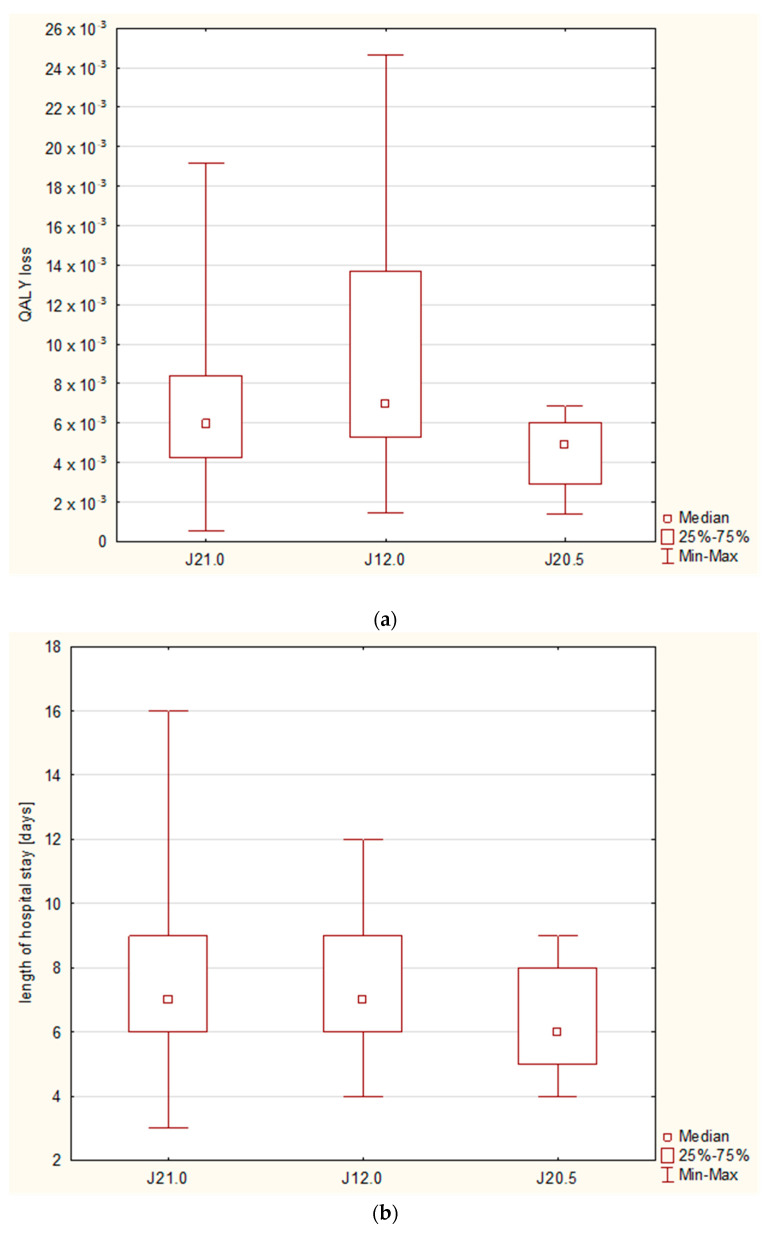
A comparison of the study subgroups regarding the final diagnosis: J21.0—bronchiolitis, J12.1—pneumonia, J20.5—bronchitis: (**a**) a box/whisker chart for the comparison of QALY loss (significant differences between bronchitis and pneumonia*) and (**b**) a box/whisker chart for the comparison of length of hospital stay (no statistically significant differences*). The results are based upon the Kruskal–Wallis test with multiple rank comparison.

**Figure 5 diseases-11-00111-f005:**
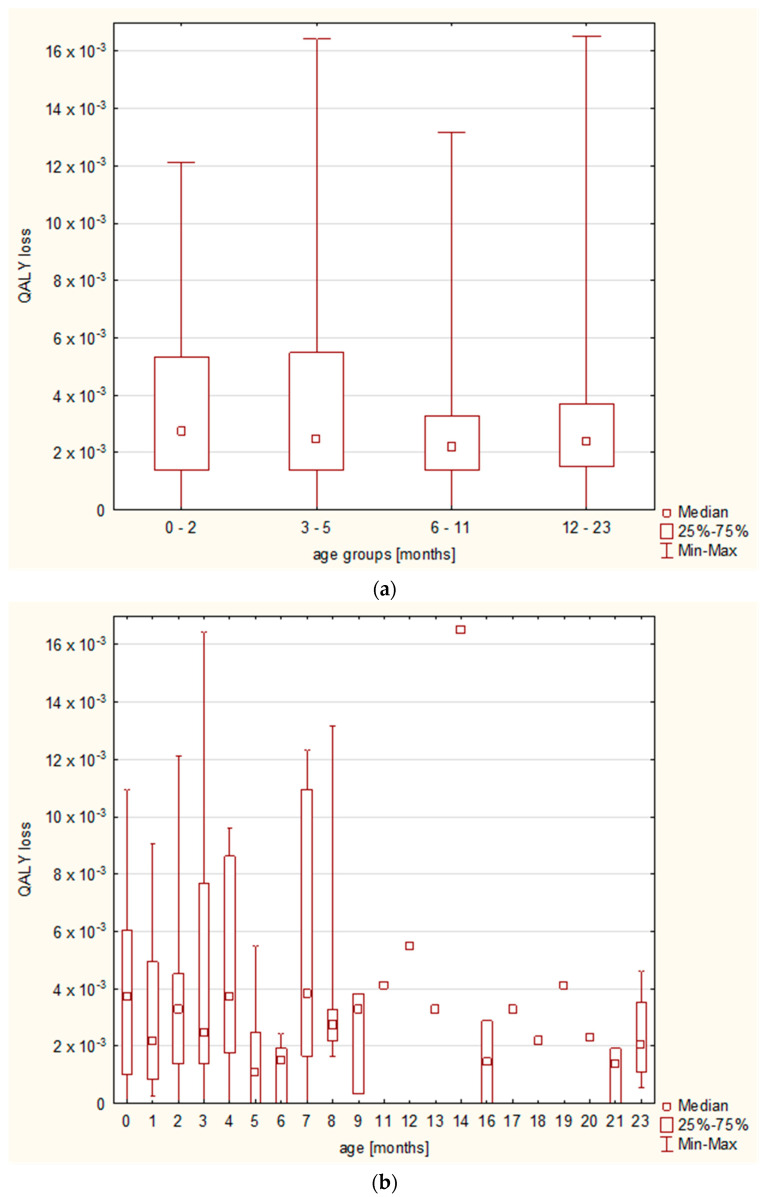
QALY losses with regard to the (**a**) age groups and (**b**) age month-by-month.

**Figure 6 diseases-11-00111-f006:**
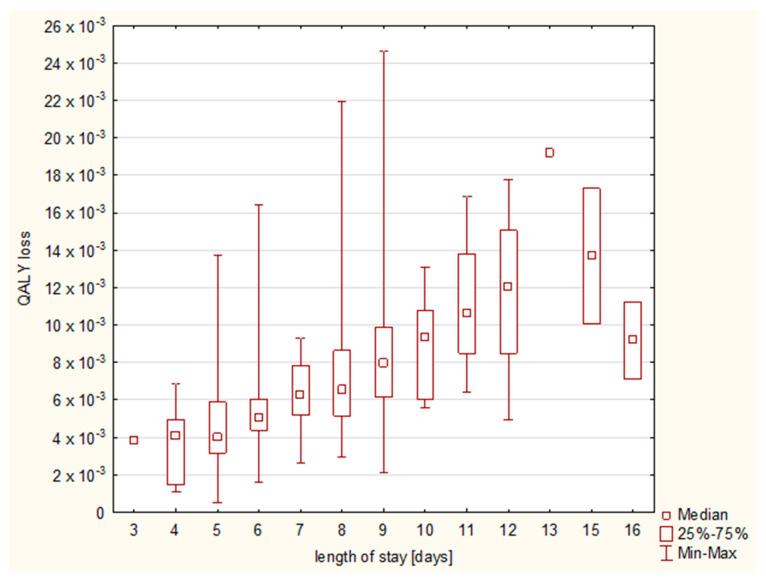
QALY loss according to the length of hospital stay.

**Figure 7 diseases-11-00111-f007:**
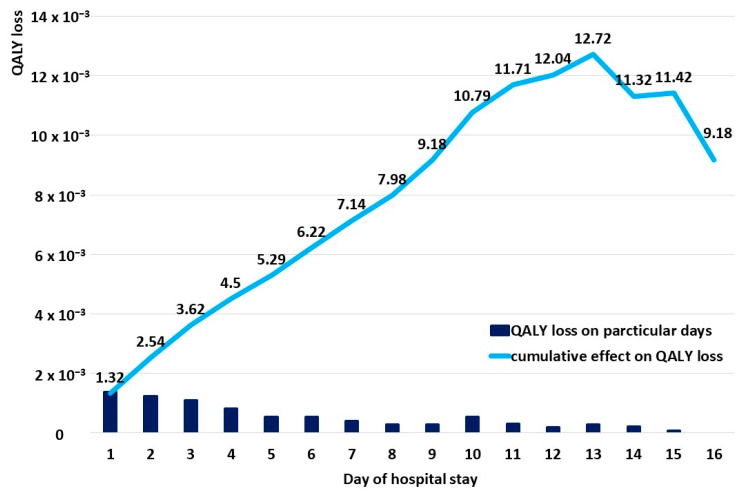
A cumulative effect of the length of hospitalization on QALY loss (solid blue line) combined with QALY loss on particular days of hospital stay (dark blue rectangles).

**Table 1 diseases-11-00111-t001:** Inclusion and exclusion criteria.

Inclusion Criteria	Exclusion Criteria
Age: 0–23 months old Presence of signs/symptoms suggestive of a respiratory tract infection (runny nose, cough, dyspnea, increased body temperature, abnormalities in physical examination according to the definitions presented below) Laboratory confirmation of the RSV infection (positive rapid antigen test or RT-PCR) Community origin of the RSV infection, i.e., signs/symptoms present prior to hospitalization or up to the first 48 h after hospital admission Informed consent Criteria for final ICD-10 diagnosis: - pneumonia (J12.1): (a) 2 or more of the following signs/symptoms: fever of ≥38 degrees Celsius, cough, intercoastal spaces retractions, tachypnoea (>60 breaths/minute in <1 month olds, >50 breaths/minute in 1–11 months olds, and >40 breaths/minute in 12–24 months olds); (b) a presence of bronchial murmur/crackles, or a dull percussion; (c) a presence in chest radiograph of consolidation(s) or parenchymal infiltrate(s) or densities (irregular/linear) or presence of pathological fluid in pleural cavity OR abnormalities consistent with pneumonia in chest ultrasound, i.e., hypoechoic pulmonary lesion(s) or hypoechoic pleural line or local absence of pleural line or area of hyperechogenicity within the consolidation or impairment of lung sliding; - bronchitis (J20.5): a presence of signs/symptoms (cough and examination wheeze or rales); - bronchiolitis (J21.0)—the first episode of bronchiolar constriction with a presence of examination wheeze or rales), dyspnoea on expiration, which may lead to hypoxia	nosocomial infection (i.e., first signs/symptoms after 48 h since hospital admission) lack of full knowledge on the clinical course of the disease (a discharge on parent’s/tutor’s request, patients transferred to another hospital) lost to follow-up (lack of a baseline QoL assessment in a disease-free period)

**Table 2 diseases-11-00111-t002:** Baseline characteristics of the included patients. Abbreviations: n—number of patients, J21.0—RSV bronchiolitis, J12.1—RSV pneumonia, J20.5—RSV bronchitis, IQR—interquartile range.

	Whole Study Group (n = 132)	J21.0 (n = 100)	J12.1 (n = 22)	J20.5 (n = 10)
male-to-female ratio	73/59 ^1^	51/49	13/9	9/1
age ^2^ median (IQR) [months]	3.0 (2.0–6.0)	3.0 (1.0–5.0)	6.0 (3.0–17.0)	14.5 (6.0–21.0)
length of stay ^3^ median (IQR) [days]	7.0 (6.0–9.0)	7.0 (6.0–9.0)	7.0 (6.0–9.0)	6.0 (5.0–8.0)
kappa coefficient median (IQR)	1.0 (1.0–1.0)	1.0 (1.0–1.0)	1.0 (1.0–1.0)	1.0 (1.0–1.0)
expected QALY * median (IQR)	365.0 (365.0–365.0)	365.0 (365.0–365.0)	365.0 (365.0–365.0)	365.0 (365.0–365.0)

^1^ *p* = 0.0565, statistically insignificant (two-way Chi-square test); ^2^ *p* < 0.01 statistically significant differences observed between J21.0–J12.1, J21.0–J20.5, and J12.1–J20.5 (Kruskal–Wallis test with multiple rank comparison); ^3^ *p* = 0.3382 (Kruskal–Wallis test), statistically insignificant; * the expected QALY calculated for a hypothetical year without RSV hospitalization and based upon the baseline kappa coefficient.

## Data Availability

Data are available upon request from the authors.

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
