# Peer review of "The Impact of RSV Hospitalization on Children’s Quality of Life"

_diseases, 2023, doi:10.3390/diseases11030111_

Round 1
Reviewer 1 Report
The manuscript appears very long-winded with a discussion that does not lead to any conclusion. it does not seem to add anything relevant to the clinical management of the infection
Author Response
Dear Reviewer,
Thank you very much for your opinion and we would have liked a bit more feedback from you. I am pleased to say that we also received a lot of constructive remarks and positive comments from 3 other reviewers, which led to the improvements in the manuscript that I thought you might like to familiarize yourself with and see our perspective.
But first, please, let me answer your comments one by one (I have italicized your comments):
The manuscript appears very long-winded with a discussion that does not lead to any conclusion.-
I totally agree that this is one of the most important skills - to express one's thoughts in a short and understandable way. In our defense, the length of the manuscript is related to the complexity of the topic, there are even more results we would like to show, but we wanted to limit the length of the paper. On the other hand, none of the data presented deserves to be in an appendix only. I have shortened some sections of the manuscript, but there are also some additions, since the reviewers were interested in deepening the topic and asked for more details or comments that I think might be of interest to you as well.
it does not seem to add anything relevant to the clinical management of the infection- Well, the study, nor the paper do not claim to add anything to the clinical management.
We are a team of clinicians and the most important part of our scientific interest is to make science responsive to the clinical management of our patients, which means to make it responsive to our needs as well. But these are not just our needs, these are the needs of the patients and their parents who are not worried about the reduced oxygen saturation in the blood or the dose of the drug, they are worried about the quality of life and the fact that a child is suffering.
Personally, I find the creation of useless "science", i.e. not relevant to clinical improvement, in areas where a clinical approach is needed to be deeply redundant and inappropriate.
Unfortunately, there is still a huge lack of information on many of the most important issues in pediatrics. I realize that this may not be the most pressing issue, but it is very important.
And while this study does not change clinical management, it does highlight one of the most underestimated and underrepresented areas of the RSV disease in publications - the impact from the patient's perspective. As pharmacoeconomic evaluations usually only include the payer's point of view, there are very few studies that focus on this practical aspect of pediatrics.
HOWEVER, in terms of clinical management, I dare say that this study could lead to an improvement in clinical management; for example, while many strategies have been shown to be ineffective in RSV bronchiolitis, for example, the studies were based on different endpoints, but the vast majority focused on improving survival or shortening hospital stay, while none of them focused on assessing QoL in children undergoing therapy. This can be a double-edged sword: while an improvement in QoL with, let’s say, a bronchodilator may be highly desirable (if observed and significant, but for this we need baseline studies that introduce the issue of QoL), other therapies may be associated with worse patient QoL - we observe that many children undergoing HFNC experience reduced QoL. In general, a change in clinical management was not the aim of the study, but this study may provide a stimulus to look at the bigger picture of the disease. Another important issue that cannot be ignored is the impact of the disease, which is also neglected - as long as we do not talk about worsened QoL, we will not see it, we will not take it into account. To summarize the discussion, we added a short paragraph on the practical implications of the study, which would make the study aims clearer.
Regarding your other ratings:
Introduction and references- since there were no negative comments about the introduction section or the references, we did not change them.
Research design- it has been changed a little bit, one the reviewers suggested performing a correlation analysis between the days and the utilities which turns out to be both a simple and a great idea- a corresponding sentence in the MM section and a chart have been added
Are the methods adequately described?- we added also a clarification on the cumulative effect calculations
Are the results clearly presented?- we improved 2 figures, and added another one, and made some minor changes
Are the conclusions supported by the results?- As mentioned before, we added a paragraph which would directly explain the meaning of the study in a broader perspective.
Of you wish, I would be pleased to provide other suggestions that we referred to, partially you can see the effects- all the corrections are in the track changes mode in order to be easily visible.
Best regards,
The authors
Reviewer 2 Report
The paper reports on the quality of life reduction of patients under 2 y.o. hospitalized for RSV, with the influence of the hospitalization time period and the day of observation. Despite the QoL issue was already quite much studied, this paper is well written and the authors provide adequate positioning with respect to these other works. However, there are still some clarifications/discussions needed on the method used and some figures must be improved. The paper can be acceptable with the following minor revisions:
1.- L134-145: was the QoL assessed by one single question as the text let’s think, or using another type of scale? The fact that the test was applied to assess very young children is an issue. Was a tool with multiple items like the Health Utilities Index? The exact test should be mentioned, and its choice discussed with relevant references.
2.- Same issue for the kappa coefficient.
3.- L158: for the QALY loss, it is considered that the loss of QoL, assessed via kappa, applies during 365-l days. This should be checked or justified, since it seems odd that the post-hospitalization impact applies for a lower time if the hospitalization was longer. I would rather expect a longer period with reduced QoL when the hospitalization is longer, since it is likely to correspond to a more severe case, (or at least equal).
4.- Figure 2 does not meet the required standards: no caption on the axes and the reader must guess that “general, 3, 4, 5,…” on the legend refer to the hospitalization duration. Fig. 5 to be checked too. For some figures, the quality should be improved.
Each point of fig. 2 is based on an average of several patients. For each hospitalization length, the figure should include the sample size.
The meaning of “cumulative effect” in fig. 5 is not clear either. To avoid any confusion, a sentence should specify how you switch from QALY loss to cumulative (can be in Method too).
5.- The perspectives are rather poor and the paper solely focuses on analyses, with no outlook into solutions. In the discussion and/or conclusion, some implications in terms of public health may be given. For instance, what should be studied in the hospitalization to start finding solutions to the QoL loss? This might increase the paper impact.
6.- Edition: in many places, “-“ was used to cut words at the end of lines, but they now appear in the middle of lines: ex. “laborato-ry”.
Author Response
Dear Reviewer!
We would like to express our gratitude for Your constructive remarks and feedback that will improve this manuscript! I would like to address directly all the issues that You raised and the minor revisions that You suggested (I italicized Your comments):
1.- L134-145: was the QoL assessed by one single question as the text let’s think, or using another type of scale? The fact that the test was applied to assess very young children is an issue. Was a tool with multiple items like the Health Utilities Index? The exact test should be mentioned, and its choice discussed with relevant references.- Well, the issue of assessing QoL in children is very complicated, and it was even more complicated at the start of this study. We are well aware (and we were at the beginning of the study) that we were trying to assess QoL in children under 2 years of age, so a proxy perspective was the only reasonable option. This is precisely why we chose a visual analogue scale, without any other tools that would assess multiple items.
Moreover, when searching in the literature, it turns out the items that should be assessed depend hugely on disease (or rather a type of disease, different aspects are affected in the case of gastroenteritis and other in the case of lower respiratory tract infection). To do justice, we did our best to find ANY tool that could be used in such a young group of patients, contacting the authors (or license holders) of the tests took us more than 6 months, and the response was always the same - the test was not suitable for the youngest group of patients. In some cases, even where the multiple test had been used in younger children, further evaluation (i.e. after the study was published) suggested that the lower age limit should be higher (e.g. EQ5D), and the newer versions were supplied with newer (higher) age limits.
While in fact we did use a single question (a caregiver's rating on the VAS) to assess the child's QoL, this procedure was repeated on a daily basis, so we certainly did not base the whole assessment on the single question, but on a series of VAS ratings in each case. The choice of VAS was thoroughly analyzed before the study was designed and at the start of the study (there have been some new ideas and even studies evaluating them, but there is still no universally accepted scale/score system for assessing QoL in the youngest patients). The use of the HUI (and any other validated scale) was out of the question due to the above-mentioned age-related limitations of its application.
Of note, analysis of changes over time is important, and for this reason we have showed in Figure 2 how utilities change in subgroups of patients (subgroups based on length of stay, we have not showed data for individual patients).
2.- Same issue for the kappa coefficient.- I need to make one clarification here before moving on to the next question. The kappa assessment was expected to correspond to the baseline QoL (also assessed by a parent/tutor) in a disease-free period, which also means a disease-sequelae-free period. The child's kappa was checked once, as it was expected to reflect the general state of health. It should be emphasized that the sequence may play a role here, as the caregivers first assessed QoL (on a 100-point scale) during hospitalization and then after discharge in a disease-free period. The kappa should therefore remain the same unless a new (temporary or permanent) condition occurs. The idea of the baseline assessment of health status (kappa coefficient) was to eliminate the influence of other health disorders and conditions, which may be significant, for example, in preterms. Of note, many of the studies assessing influence of short-term conditions (like infectious diseases) did not take into account general health status of a patient, but compared preterms versus no-preterms for example, so we dare to consider kappa verification as an advantage of our study.
3.- L158: for the QALY loss, it is considered that the loss of QoL, assessed via kappa, applies during 365-l days. This should be checked or justified, since it seems odd that the post-hospitalization impact applies for a lower time if the hospitalization was longer. I would rather expect a longer period with reduced QoL when the hospitalization is longer, since it is likely to correspond to a more severe case, (or at least equal).- I totally agree that a longer hospitalization would affect the patient more and would result in lowered QoL for a longer period of time. However, kappa is a general baseline health status of a patient and if kappa is lowered this would mean that for the whole year (minus the period of the RSV hospitalization) QoL will be affected to the same extent due to, let’s say, prematurity.
In addition, when there is a trigger (RSV hospitalization) other than a chronic condition, QoL is lowered, but this should not be seen as 100% QoL in every case (although the vast majority of caregivers reported that their children had 100% QoL), but should be assessed in relation to the baseline health status (for example, a loss of utility of 70% is not the same reduction for a completely healthy person as for a person with a chronic condition who assesses their QoL as 75% every day outside the RSV hospitalization). So kappa is supposed to reflect general QoL of a child in a disease-free period.
While kappa is NOT a POST-HOSPITALISATION impact, it only assesses a patient's daily QoL, there are still 2 possibilities of error: 1) the post-hospitalization impact of RSV is higher than the QoL deterioration observed during hospitalization (highly unlikely, but possible); 2) the duration of the RSV-related QoL deterioration is longer than expected (we discussed this point briefly), and in the expected disease-free period we actually meet the patient still suffering from RSV disease (however, here the question "Are you currently observing any signs/symptoms in your child?" was asked to check whether the timing of the kappa assessment was appropriate).
4.- Figure 2 does not meet the required standards: no caption on the axes and the reader must guess that “general, 3, 4, 5,…” on the legend refer to the hospitalization duration. Fig. 5 to be checked too. For some figures, the quality should be improved.- I am sorry! I corrected the Figure, adding caption of the axes, I changed the somehow misleading name of “general” into “overall median”.
Each point of fig. 2 is based on an average of several patients. For each hospitalization length, the figure should include the sample size.- Definitely! I added the number of patients per each subgroup. Thank You for this comment, this add transparency to the results and makes the reading easier.
The meaning of “cumulative effect” in fig. 5 is not clear either. To avoid any confusion, a sentence should specify how you switch from QALY loss to cumulative (can be in Method too).- Definitely! In brief, the cumulative effect was created by adding QALY loss on particular days, but again data distribution was verified and presented in adequate manner. I added a corresponding sentence in the MM section.
5.- The perspectives are rather poor and the paper solely focuses on analyses, with no outlook into solutions. In the discussion and/or conclusion, some implications in terms of public health may be given. For instance, what should be studied in the hospitalization to start finding solutions to the QoL loss? This might increase the paper impact.- Thank You for this comment, this is also a great idea! On one hand we did not intend to make the discussion too long, but this paragraph would definitely increase the value and the MEANING of the paper- we added a short comment on the topic in order to underline the message.
6.- Edition: in many places, “-“ was used to cut words at the end of lines, but they now appear in the middle of lines: ex. “laborato-ry”. The issue with the dashes was probably due to some kind of coding error, I corrected them (I hope I have corrected all of them).
Best regards,
The authors
PS I would like to take this moment to express my personal gratitude for such a substantive comment, which to be honest doesn't happen too often in the review process at the moment.
Reviewer 3 Report
The manuscript is well-written, succinct and describes results of significant general interest. The methods appear to be described carefully and in sufficient detail for the work to be repeated by others if necessary. The standard of presentation is generally high. As a structural biologist I am not able to judge the statistical analysis but as far as I can tell, the paper would seem to merit publication as is, providing that the opinion of an expert medical statistician is sought.
Author Response
Dear Reviewer, thank you very much for your kind review and opinion!
Regarding the statistical analysis that You mention, we paid a lot of attention to this part of our publication, as such a complicated topic deserves a decent statistical analysis, so we would like to assure You that it was done carefully. At the request of one of the reviewers, we have done an additional correlation analysis and added a graph, which together reflect the results more thoroughly.
Best regards,
The authors
Reviewer 4 Report
The article Diseases-2538870 examined quality of life judgments made by caregivers of children hospitalized for RSV virus infections.
The quality of English is quite strong throughout, but could be improved by a minor edit. Two sample problems are the inclusion of dashes (-) in inappropriate palces (e.g., 12, 48, 49) and the use of "showed" instead of "shown".
The basic data of this research were parent ratings. The authors should further discuss issues of validity and reliability with such data. Small factors such as time-of-completion for the VAS scale might influence results, especially if the caregiver had just seen/not seen a physician. As well, the distance between hospital stay and the normative rating might influence ratings.
The exclusion of "nosocomial" cases is interesting: one is not sure if the disease had its origin within the hospital, but this exclusion was probably a good idea.
The testing for RSV, and the process of disease categorizatio following ICD were both well described.
The spacing in Table 1 is awkward.
The authors discuss many formulas for utility loss, but mostly analyze utility. Too much is going on here. Perhaps they should stick to two or three measures, explaining why these measures are useful to the research.
Many of the descriptive tables seem to contribute mainly noise to the article - most especially the box plots. However, if these are likely to be of interest to readers who appreciate that level of detail, they can be maintained.
Figure 2 needs a better explanation for what is being plotted. The axes of the graphs need to be labeled. The blue lines should probably be called "overall average" (if this is what they were), and line colors should clearly be labelled as showing length of stay.
Additionally, the data in Fig 2 can be treated statistically - separately for each line - where day of stay is correlated with utility (r or rho). This would support the authors discussion of increasing utility across stay.
Figure 5 confounds several variables. The shape of the line might be due to the fact that only those with lower utility remain for many days, so when only they are left in the graph the line dips. Having carefully avoided this problem by producing Figure 2, I do not think the authors should use Figure 5.
The authors discuss their results very carefully in comparison to the existing literature. This is a strong point.
Only minor editing is needed here.
Author Response
Dear Reviewer,
Thank you very much for your thorough review and all the comments and tips, that, we believe, will improve our paper significantly. We would like to report point-by-point our improvements according to your comments (I italicized You comments):
The quality of English is quite strong throughout, but could be improved by a minor edit. Two sample problems are the inclusion of dashes (-) in inappropriate palces (e.g., 12, 48, 49) and the use of "showed" instead of "shown"- Thank you for the comment, we did our best to make the English at least readable. The problem with the dashes was probably due to some kind of coding error, I have corrected it (I hope I have corrected them all). As for "showed" and "shown", this was written in American English, so I left it as it was.
The basic data of this research were parent ratings. The authors should further discuss issues of validity and reliability with such data. Small factors such as time-of-completion for the VAS scale might influence results, especially if the caregiver had just seen/not seen a physician. As well, the distance between hospital stay and the normative rating might influence ratings.- Certainly, many factors could have influenced the assessment, we gave a short comment on it in the manuscript, would love to discuss the issue further, however, we also got the opinions on too long and too detailed discussion... Regarding the time of completion, it is very interesting topic, however, an effort has been made not to influence parental judgement, and all the assessments were made irrespective or physicians' visits at the end of the day. We had an interesting discussion during the Ethics Committee meeting and some other points that might influence the result were also suggested, like personal situation, level of support by family, or even the number of persons per room or the behavior of hospital room-mates. I totally agree with this point, however, no further distinction could have been made, since they would be too detailed for this number of patients. I added a comment on this topic into study limitations.
The exclusion of "nosocomial" cases is interesting: one is not sure if the disease had its origin within the hospital, but this exclusion was probably a good idea.- This is another idea of factors potentially influencing the rating; we excluded patients with nosocomial origin of the disease, since they have already stayed for a period of time, and a prolonged hospitalization may be more challenging and overestimate true RSV effects. We hope this was a good idea.
The testing for RSV, and the process of disease categorizatio following ICD were both well described. Thank you!
The spacing in Table 1 is awkward.- In fact, it did not look good. I changed the whole table, hope it is more transparent now.
The authors discuss many formulas for utility loss, but mostly analyze utility. Too much is going on here. Perhaps they should stick to two or three measures, explaining why these measures are useful to the research.-
To be honest, I am not 100% sure which part you are referring to. I hope we did not overwhelm the reader in the introduction section, but in the MM section we had to show the detailed formulas for the calculations, we were also asked in the review process to provide information on cumulative effect calculations. Regarding the study design, it is true that we chose different measures, but I think I can explain why they are all necessary and do not overlap, although they are based on each other. In fact, we used utility assessment (with VAS) and QALY calculations with all the analyses of their changes - relationship with age or diagnosis and changes over time. While utility and utility loss are easily transformed into each other, they can serve as a basis for QALY calculations, but QALYs depend on QoL in a disease-free period. In other words, it depends on the utilities in a disease-free period. However, as we did not assess utility outside the RSV hospitalization (there was only a kappa assessment, which is something of a novelty compared to other studies), we could only use the QALY calculation (and an assessment of the RSV-attributable QALY loss) to present the results in a broader perspective. Another advantage of the QALY is that it can be used globally, provides data to calculate ICER for different interventions, and is widely accepted. However, many of the studies reported so far have provided utility calculations, and not all of them have used QALYs. In the discussion section, we took the liberty of discussing the topic thoroughly, hoping not to exaggerate.
Many of the descriptive tables seem to contribute mainly noise to the article - most especially the box plots. However, if these are likely to be of interest to readers who appreciate that level of detail, they can be maintained.- Actually, we encountered some problems when referring to the results in this manuscript, as they lacked some minor results; precisely for this reason we allowed ourselves to show the tables and charts- just in case someone was interested in the topic or needed more detailed data;.
Figure 2 needs a better explanation for what is being plotted. The axes of the graphs need to be labeled. The blue lines should probably be called "overall average" (if this is what they were), and line colors should clearly be labelled as showing length of stay.- This is a great idea, thank you! I changed the “general” into “overall median” (due to data distribution we presented here median not average), but that definitely reflects the issue more accurately! I also added labels to the graphs and changed the figure caption.
Additionally, the data in Fig 2 can be treated statistically - separately for each line - where day of stay is correlated with utility (r or rho). This would support the authors discussion of increasing utility across stay.- That is a brilliant idea!!! I haven’t even thought of it earlier- a median utility was calculated for each day and with the use of Spearmann’s rank compiled against the day of stay, irrespective of the hospitalization length groups. It turns out… there is 93.5% correlation! Thank You for so much for this point, it is awesome! I added also a chart.
Figure 5 confounds several variables. The shape of the line might be due to the fact that only those with lower utility remain for many days, so when only they are left in the graph the line dips. Having carefully avoided this problem by producing Figure 2, I do not think the authors should use Figure 5.- In fact, the observed drop in a cumulative QALY loss can be caused by the fact that those with lower utility LOSS remained for a longer period of time. We are aware of this fact and of its practical implications, and literally addressed this issue in the discussion section (“Thus, individual characteristics of the patients may have had impact on the less expressed cumulative effect, since those hospitalized for a longer period of time may not have exhibited such a high total QALY loss; however, the graph shows that in the case of our patients, the QALY loss lasted at least until day 15”). We would love to give more comments on this topic, however, taken into account the length of the discussions, that might be problematic. While Fig. 2 refers to utility loss, Fig. 5 refers to QALY loss, which is not exactly the same, that is why I left the Figure 5 (now renumberded as Fig.6 due to the addition of Spearman’s rank correlation chart) as well.
The authors discuss their results very carefully in comparison to the existing literature. This is a strong point.- Thank You very much, I believe this topic deserves lots of attention and should be presented and discussed thoroughly. We added also a short paragraph on practical implications of this studies and future directions.
Best regards,
The authors